# Gene Set Enrichment Analysis Reveals That Fucoidan Induces Type I IFN Pathways in BMDC

**DOI:** 10.3390/nu14112242

**Published:** 2022-05-27

**Authors:** Suyoung Choi, Sol A Jeon, Bu Yeon Heo, Ju-Gyeong Kang, Yunju Jung, Pham Thi Thuy Duong, Ik-Chan Song, Jeong-Hwan Kim, Seon-Young Kim, Jaeyul Kwon

**Affiliations:** 1Department of Medical Science, College of Medicine, Chungnam National University, Daejeon 35015, Korea; jdd02287@naver.com (S.C.); xeyk1603@naver.com (B.Y.H.); yunjuJudithJung@gmail.com (Y.J.); phamduong290194@gmail.com (P.T.T.D.); petrosong@cnuh.co.kr (I.-C.S.); 2Department of Infection Biology, College of Medicine, Chungnam National University, Daejeon 35015, Korea; 3Brain Korea 21 FOUR Project for Medical Science, Chungnam National University, Daejeon 35015, Korea; 4Personalized Genomic Medicine Research Center, Korea Research Institute of Bioscience and Biotechnology, Daejeon 34141, Korea; jsa9385@kribb.re.kr (S.A.J.); alternar@kribb.re.kr (J.-H.K.); 5Department of Bioscience, University of Science and Technology, Daejeon 34113, Korea; 6Department of Biological Sciences, Korea Advanced Institute of Science and Technology, Daejeon 34141, Korea; kangju7@kaist.ac.kr; 7Department of Internal Medicine, Chungnam National University, Daejeon 35015, Korea; 8Department of Medical Education, College of Medicine, Chungnam National University, Daejeon 35015, Korea; 9Translational Immunology Institute, Chungnam National University, Daejeon 35015, Korea

**Keywords:** BMDC, dendritic cells, fucoidan, differentially expressed genes (DEGs), gene set enrichment analysis (GSEA), type I IFN, innate immune cells, PRR, SARS-CoV-2, antiviral

## Abstract

Fucoidan, a sulfated polysaccharide extracted from brown seaweed, has been proposed to effectively treat and prevent various viral infections. However, the mechanisms behind its antiviral activity are not completely understood. We investigate here the global transcriptional changes in bone marrow-derived dendritic cells (BMDCs) using RNA-Seq technology. Through both analysis of differentially expressed genes (DEG) and gene set enrichment analysis (GSEA), we found that fucoidan-treated BMDCs were enriched in virus-specific response pathways, including that of SARS-CoV-2, as well as pathways associated with nucleic acid-sensing receptors (RLR, TLR, NLR, STING), and type I interferon (IFN) production. We show that these transcriptome changes are driven by well-known regulators of the inflammatory response against viruses, including IRF, NF-κB, and STAT family transcription factors. Furthermore, 435 of the 950 upregulated DEGs are classified as type I IFN-stimulated genes (ISGs). Flow cytometric analysis additionally showed that fucoidan increased MHCII, CD80, and CD40 surface markers in BMDCs, indicative of greater antigen presentation and co-stimulation functionality. Our current study suggests that fucoidan transcriptionally activates PRR signaling, type I IFN production and signaling, ISGs production, and DC maturation, highlighting a potential mechanism of fucoidan-induced antiviral activity.

## 1. Introduction

Fucoidan is a sulfated polysaccharide found in various species of brown seaweed. It is largely composed of L-fucose and sulfate ester groups, but can contain other monosaccharides (mannose, galactose, glucose, xylose, etc.), and uronic acids depending on the extraction process and seaweed species [1]. Fucoidan has been reported to possess a variety of biological activities, including antiviral, anti-inflammatory, anticoagulant, and antitumor effects, making it an attractive target for the development of nutraceuticals and functional foods [2,3,4,5,6]. While fucoidan has garnered particular interest for its antiviral properties, further investigation is required to understand its underlying molecular and cellular mechanisms.

The antiviral effects of fucoidan have been explored in many different in vitro and in vivo infection models, including that of herpes simplex virus (HSV) [7,8,9,10], human immunodeficiency virus (HIV) [11,12,13], human T-lymphotropic virus type-1 (HTLV-1) [14], avian influenza virus (AIV) [15], human cytomegalovirus (HCMV) [16], Newcastle disease virus (NDV) [17,18], influenza virus [19,20,21], and hepatitis B virus [22,23]. In recent reports it has been suggested that fucoidan and other sulfated polysaccharides from seaweed have a prophylactic and therapeutic effect against SARS-CoV-2 infection [24,25,26,27]. While various mechanisms specific to SARS-CoV-2 have been proposed, including binding of the spike glycoprotein to prevent cell entry [24,28], how fucoidan exerts its effects on such a wide variety of viruses remains unknown.

Rapid and robust synthesis of type I IFNs constitute the primary protective response to most viral infections [29]. Specifically, IFNs activate the JAK-STAT pathway to drive the expression of hundreds of IFN-stimulated genes (ISGs), which in turn directly and indirectly exert antiviral effects, as well as promote antigen presentation and adaptive immunity activation [6,30]. Accumulating evidence indicates that SARS-CoV-2 targets the host’s type I interferon (IFN) response at multiple steps, including viral sensing and IFN signaling [31,32,33]. Viruses are sensed by pattern recognition receptors (PRRs) such as toll-like receptors (TLRs), nucleotide-binding oligomerization domain-like receptors (NLRs), retinoic acid–inducible gene-I–like receptors (RLRs), and C-type lectin receptors (CLRs) [34,35]. By inhibiting PRR signaling, viruses such as SARS-CoV-2 may evade detection and attenuate IFN production [32,36,37]. They may also directly inhibit IFN production and signaling, as suggested by patients with mild coronavirus disease 2019 (COVID-19) demonstrating early and strong IFN responses whereas those with severe disease showing a deficiency in IFN synthesis [38,39].

One of the primary cells indicated in early viral detection and IFN production are dendritic cells [40,41]. Plasmacytoid DCs (pDCs) in particular are able to produce large amounts of type I IFN in response to viral infection, far exceeding those produced by other cells [42]. Indeed, patients with severe COVID-19 displayed low levels of circulating pDCs with a pro-inflammatory phenotype, suggesting a diminished ability to produce type I IFN [42,43,44,45,46,47]. In addition, DCs bridge the innate and adaptive antiviral immune responses via their role as professional antigen presenting cells (APCs). Specifically, conventional DCs (cDCs) capture viral pathogens for presentation to naïve T cells, inducing cytotoxic T-cell activity and antibody production that are critical to virus eradication [48,49]. cDCs were similarly reported to express decreased levels of MHC-II in COVID-19 patients, which would lead to impaired adaptive immune system activation [50,51].

In the present study, we analyzed the effects of fucoidan treatment on the transcriptome of bone-marrow derived DCs (BMDCs) given the importance of DCs in the antiviral response. Our findings illustrate that fucoidan promotes viral sensing, IFN signaling, and adaptive immune activation in BMDCs, highlighting its potential for further investigation against SARS-CoV-2 in in vivo studies.

## 2. Materials and Methods 

### 2.1. Bone Marrow-Derived Dendritic Cell (BMDC) Culture 

Bone marrow cells were obtained from the femur and tibia of 6- to 10-week-old female C57BL/6 mice (Damulscience, Korea) and differentiated to dendritic cells as previously described [52]. Briefly, bone marrow cells were cultured in RPMI medium containing 10% FBS, 1% penicillin/streptomycin, 50 μM 2-Mercaptoethanol, and 20 ng/mL GM-CSF for 3 days. The media was replaced with fresh supplemented media and cultured for an additional 3 days. Nonadherent immature dendritic cells were harvested on day 6, treated with 400 μg /mL fucoidan (20–200 kDa) (F8190, from *Fucus vesiculosus*, Sigma-Aldrich, St. Louis, MO, USA) [53,54] or vehicle (0.04% DMSO) in complete media for 24 h, and subjected to RNA extraction for RNA-seq analysis. For flow cytometry analysis, immature dendritic cells were treated with 100 to 400 μg /mL fucoidan or vehicle in complete media for 24 h.

### 2.2. Cell Staining and Flow Cytometry

Cells were stained with the following antibodies for 30 min at 4 °C and analyzed using a flow cytometer (FACS Canto II, BD Biosciences, San Jose, CA, USA) and FlowJo software (Tree Star, Ashland, OR, USA): PerCP-Cy5.5-MHCⅡ (562363, BD Biosciences, San Jose, CA, USA), PE-Cy7-anti-CD80 (25-0801-82, eBioscience San Diego, CA, USA), APC-anti-CD11c (20-0114-U100, Tonbo bioscience, San Diego, CA, USA), APC-Cy7-CD11b (557657, BD Biosciences, San Jose, CA, USA). Cell viability was measured by LIVE/DEAD Fixable Violet dye (L34955, Thermo Fisher Scientific, Waltham, MA, USA) Mouse Fc receptors were blocked with Mouse Fc Block™ (553142, BD Biosciences, San Jose, CA, USA) for 15 min at 4 °C.

### 2.3. RNA Sequencing

Total RNA was isolated using the Trizol reagent (Thermo) and RNeasy kit (Qiagen) and the purified mRNAs were used to synthesize double-stranded cDNAs using a SuperScript Double-Stranded cDNA Synthesis Kit (Thermo Fisher Scientific). Sequencing libraries were constructed using a NEBNext Ultra II DNA Library Prep Kit for Illumina (NEB) and sequenced on the Illumina HiSeq 2500. Raw fastq files were quality-checked with FastQC (v0.11.9), and subjected to trimming with ‘Trim galore!’ (v0.6.4) of the Python tool (FastQC score Q30 over) to remove adapters and low-quality reads. For subsequent mapping, mouse reference genome (mm10) data were obtained from Illumina iGenomes (https://sapac.support.illumina.com/sequencing/sequencing_software/igenome.html). Sequences were then aligned to the mouse reference genome (mm10) with STAR (v2.7.3a). For the mapped data, low read counts and duplicate reads were removed, and the total read count, counts per million (cpm), and log FC were obtained.

### 2.4. Bioinformatic Analyses

#### 2.4.1. Analysis of Differentially Expressed Genes in IDEP

Gene expression read counts were exported and analyzed through the integrated Differential Expression and Pathway (iDEP) tool (http://bioinformatics.sdstate.edu/idep93/). An MA plot was created to identify differentially expressed genes (DEGs) using the DESeq2 method with a threshold false discovery rate (FDR) < 0.1 and fold-change > 2. To characterize the biological function of the DEGs, KEGG pathway enrichment analysis was performed and visualized on a hierarchical tree and network. In addition, TF motifs enriched in gene promoters defined as 300 bp upstream of DEGs were identified and visualized as a hierarchical tree and network.

#### 2.4.2. GSEA

The Broad Institute’s Gene Set Enrichment Analysis (GSEA) software (version 4.1.0) was used with the C2 gene sets in the Molecular Signatures Database (MSigDB). A normalized enrichment score (NES) was also calculated by GSEA in which differences in pathway size (i.e., gene set size) were considered, allowing for comparisons between pathways within the analysis. This ES reflects the degree to which a gene set is overrepresented at the top or bottom of a ranked list of genes. Positive and negative NES values represent enrichment at the top and bottom of the list, respectively. The analysis involved 1000 gene set permutations with gene sets limited to 15–500. Initially, a FDR of 25% was used for all bioinformatics analyses.

Leading-edge analysis was performed after each GSEA to determine the core genes that have the highest impact on the biological process of a given gene-set. GSEA was run on a database of C2, and all canonical pathways and the enriched gene sets in KEGG, Reactome, and WikiPathways (WP) (*p* < 0.05; FDR < 0.25) were selected for leading-edge analysis. 

To understand the role of TFs in pathway perturbations after fucoidan treatment, TF enrichment analysis was performed with GSEA using the TFT_Legacy subset of C3: regulatory target gene sets (*p* < 0.05; FDR < 0.25). In addition, GSEA leading-edge analysis of the enriched gene sets of potential TF targets were performed.

#### 2.4.3. Functional Enrichment Visualization and Annotation of GSEA Results

GSEA was performed using the ‘CP: Canonical Pathway’ gene sets in MSigDB and the GSEA results were visualized using the EnrichmentMap (version 3.3.2) module for Cytoscape (version 3.8.2). The enrichment results were mapped as a network of gene-sets (nodes) where the nodes represent statistically significant terms (Node cutoff, Q-value 0.08) and the links (edges) represent the degree of gene-set similarity (Edge cutoff, 0.25). For gene-sets annotations, the enriched gene-sets were grouped by AutoAnnotate software (version 1.3.4, http://apps.cytoscape.org/apps/autoannotate) according to the Markov cluster (MCL) algorithm based on the edge weights of similarity coefficients and were automatically annotated using the WordCloud algorithm with a maximum of three words per label. Nodes are colored by enrichment score (ES), and edges are sized on the basis of the number of genes shared by the connected pathways.

#### 2.4.4. Interferome Analysis

In order to identify whether interferon stimulated genes (ISGs) were regulated by fucoidan treatment, Interferome analysis of DEGs was performed using the IFN regulated genes (IRGs) database, Interferome (v2.01) (http://www.interferome.org/interferome/home.jspx).

### 2.5. Statistical Analysis 

Unless otherwise specified, data are shown as mean ± SEM and each experiment was repeated three times. Data were analyzed by the two tailed unpaired *t*-test or one-way ANOVA with Dunnett’s post hoc analysis using GraphPad Prism (v7.02, GraphPad). 

## 3. Results

### 3.1. Fucoidan Treatment Upregulates Gene Sets Associated with Immune Responses to Exogenous Stimuli

To investigate whether fucoidan can induce changes to the antigen-presenting cell transcriptome, mouse BMDCs were grown in GM-CSF-contained medium for 6 days and treated with fucoidan or vehicle (0.04% DMSO) for 24 h. RNA was isolated from the harvested cells (*n* = 3) and cDNA libraries were generated and sequenced for transcriptome analysis (Figure 1). After raw RNA-seq data from BMDC samples were processed to define a ranked gene list, DNA sequence reads were quality filtered by trimming to remove low-quality bases and mapped to a genome-wide reference set of transcripts to enable counting of reads per transcript. Read counts were then aggregated at the gene level (counts per gene) and counts per million (cpm), and log FC values were obtained for bioinformatics analyses. 

We began our analysis by generating a list of differentially expressed genes (DEGs) between fucoidan- and control-treated BMDCs using iDEP, a web-based tool designed for the analysis and interpretation of omics data [55]. With the DESeq2 package, we identified 950 upregulated and 537 downregulated genes using a false discovery rate (FDR) < 0.1 and fold-change > 2 (Figure 2A). The MA plot demonstrates that fucoidan treatment dramatically upregulated several genes involved in immune function, including Irf7, Il12b, H2-K1, Hmox1, and Ifit3 (Figure 2B).

To investigate biological themes among our list of DEGs, we used the DEG2 function in iDEP to identify enriched KEGG pathways (Table 1). We found that downregulated genes were strongly involved in cell cycle and DNA replication, suggesting that fucoidan treatment inhibited cell proliferation. In contrast, upregulated genes were related to pathogen recognition and signaling pathways, including C-type lectin receptor (CLR), Toll-like receptor (TLR), and NOD-like receptor (NLR) pathways. As expected, downstream cytokine and IFN signaling pathways such cytokine–cytokine receptor interaction, TNF signaling, JAK-STAT signaling, and virus-specific responses were also overrepresented. Other enriched pathways included NF-κB signaling, cancer, and antigen processing and presentation.

We then constructed a hierarchical clustering tree (Appendix A) and network (Figure 2C) in iDEP to summarize the overlap among enriched KEGG pathways [55,56]. Within upregulated genes, viral infection pathways were closely associated, including that of measles, influenza A, Epstein–Barr virus, and Kaposi sarcoma-associated herpes virus. These pathways were further associated with pathogen recognition and response pathways related to CLR, NLR, TLR, and JAK-STAT signaling. Interestingly, pathways related to adaptive immune responses such as antigen processing and presentation and type I diabetes mellitus pathways formed a separate cluster.

We then examined whether gene expression changes in fucoidan-treated BMDCs were driven by specific transcription factors (TFs). Target genes of specific TFs that were overrepresented in our list of DEGs (Table 2 and Appendix A) were visualized as a hierarchical clustering tree and network (Appendix A). Among downregulated genes, target genes of TFs Mcm 2, Mcm3, Mcm7, Mcm4, and Mcm6 were overrepresented (Appendix A). On the other hand, upregulated DEGs were enriched with target genes of the IRF, STAT and NF-κB TF families (Table 2). These target genes tended to be known regulators of immune function, including cytokines, chemokines, PI3K/AKT/mTOR pathway, cytoplasmic RNA sensors, costimulatory molecules, MHC, and antigen presentation [57,58,59].

### 3.2. Gene Set Enrichment Analysis (GSEA) Showed That Fucoidan Strongly Induces Interferon Signaling Pathways in BMDCs

Given our above findings suggesting that fucoidan induces transcriptome changes to promote immune surveillance and signaling in BMDCs, we sought to expand our analysis and generate a ranked list of genes without prior filtering by *p*-value and fold change thresholds. Mapped RNA-seq data were normalized as counts per million (cpm) using the edgeR package in R. The resulting expression data were analyzed with the Broad Institute’s gene set enrichment analysis (GSEA) software using the C2 CP (canonical pathways) sets from the MSigDB gene database [60]. GSEA of normalized counts of all expressed genes showed that 728 gene sets were significantly enriched (FDR < 25%) by fucoidan treatment, while 551 gene sets were significantly enriched in control BMDC samples. Even under fairly stringent criteria (*p*-value < 0.01), there were 402 gene sets deemed significantly enriched in fucoidan-treated BMDCs and 359 gene sets in control BMDCs.

We then applied EnrichmentMap clustering and annotation to collapse redundant gene sets and generate a network visualization (Figure 3). Like the network function in iDEP, the EnrichmentMap application groups similar gene sets (nodes) based on the number of overlapping genes [61]. We found that the largest cluster in fucoidan-treated BMDCs was ‘induction of interferon signaling’ (52 gene sets) followed by ‘ECM collagen crosslinking’ (20), ‘activation of PI3K FGFR2’ (20), ‘NRF2 transcriptional activity’ (4), ‘tetrasaccharide linker sequence’ (4), ‘therapy induced HIF1 pathway’ (3), ‘chemokine and chemokine receptors’ (2), and ‘scavenger scavenging ligands’ (2). Downregulated gene sets included that of DNA strand repair, citrate cycle, and oxidative phosphorylation.

Given our EnrichmentMap clusters, we looked for specific pathways associated with interferon signaling such as downstream antiviral effectors and upstream pathogen recognition. Similar to our findings with iDEP, fucoidan-treated BMDCs were enriched in virus-specific response pathways including that of SARS-CoV-2, respiratory syncytial virus, parainfluenza virus 3, influenza A, HCMV, KSHV, oncolytic virus, adenovirus, reovirus, MERS-CoV, and Ebola virus (Appendix A). Interestingly, RLR, TLR, NLR, and STING pathways, all of which involve nucleic acid-sensing PRRs, were also enriched (Appendix A). Other upregulated PRR pathways include that of dectin-1, class A scavenger receptors (SRs), and complement receptor 3 (CR3), which are consistent with previous reports of fucoidan-dependent signaling [62]. Altogether, fucoidan treatment stimulates the expression of gene sets related to nucleic acid-sensing, IFN induction, and virus-specific immunity in BMDCs, suggestive of an antiviral phenotype.

### 3.3. GSEA Leading-Edge Analysis Identified a Cluster of Type I IFN Signaling-Centered Host Defense Mechanisms

Within an enriched gene set, there is often a core group of genes that drives the enrichment signal and thus is thought to have the greatest phenotypic impact. To determine these gene subsets in fucoidan-treated BMDCs, C2 canonical pathway gene sets derived from the KEGG, Reactome, and WikiPathways (WP) databases were used to run a GSEA (*p* < 0.05; FDR < 0.25) for a subsequent leading-edge analysis. The resulting set-to-set diagrams are shown in Figure 4, where the intensity of a cell color demonstrates the number of shared leading-edge genes between the two gene sets. Gene sets were manually grouped into B1–B9 based on the presence of overlapping leading genes (as indicated by cell color intensity) and signaling themes (Figure 4A). The B1 group consists of gene sets related to TLR, type I IFN, IL1R, RLR, and NFκB signaling, while the B2 group includes type I IFN, IFNs, and SARS-CoV-2 infection signaling. Other notable leading genes include NRF2 and scavenger receptor signaling pathways. 

In terms of color intensity and biologic function, groups B1 and B2 may also be grouped together as one large cluster A1. Within cluster A1, various PRR signaling pathways such as RIG-I-like receptors (RLRs), Toll-like receptors (TLRs), NOD-like receptors (NLRs), and STING pathways are overrepresented (Table 3). Type I IFN, IFNs, IL1, and NFκB signaling pathways are also included in this cluster, as well as several SARS coronavirus-relevant innate immune pathways. These findings further support that fucoidan treatment may activate host defense pathways against viral pathogens, in which type I IFN signaling plays a critical role.

Among the top ten enriched gene sets in cluster A1, representative enrichment plots are shown in Figure 4B with their leading-edge genes listed in Appendix A. Notably, the enrichment score of the interferon a/b signaling Reactome gene set (Figure 4B, left) appeared to be driven by genes related to RNA-sensing pathways, including endosomal TLRs, MDA5, RIG-I, and PKR. Similarly, gene sets specific to anti-SARS-CoV-2 activity (Figure 4B, middle and right) appeared to be driven by type I IFNs, IFN receptors, JAK-STAT signaling, and several interferon-stimulated genes (ISGs). Additional leading-edge analysis of cluster A1 generated a list of shared leading-edge genes based on the number of pathways that share this gene (Table 4). Consistent with previous findings, type I IFN pathways (Ifnb1, Ifna4, Ifnar2, Ifnar1), JAK-STAT signaling (Stat1, Stat2, Jak1), and ISG genes were highly shared among enriched pathways, as well as those of NF-κB, MAPK, and PI3K-AKT pathways. Genes of DC costimulatory molecules (Cd86, Cd80, Cd40), inflammatory cytokines, and chemokines were also observed. Set-to-set and heatmap diagrams are shown in Appendix A. These results further support that viral recognition and type I interferon induction pathways are strongly induced in fucoidan-treated BMDCs.

GSEA was then performed using the regulatory target gene (TFT) set in C3 to identify TFs specifically linked to fucoidan-induced gene expression changes. Unsupervised ranking for confidence of enrichment identified IRF, NF-κB and STAT targets as among the most significant TF-related changes between samples (Figure 4C). As expected, the E2F TF family was enriched in control BMDCs (Appendix A). 

Based on our results suggesting the importance of IFN-stimulation, we analyzed DEGs previously generated in iDEP via Interferome v2.01 [61], a database of IFN-regulated genes. Interestingly, 435 of the 950 upregulated genes belonged to type I IFN-stimulated ISGs (Figure 4D), further supporting that fucoidan treatment leads to upregulation of ISGs in BMDCs. 

### 3.4. Fucoidan Stimulation Promotes BMDC Maturation and Activation In Vitro

Type I IFN signaling is known to promote DC maturation and thereby stimulate activation of the adaptive immune response [63]. Consistent with our findings that fucoidan induces strong type I IFN induction in BMDCs, GSEA of the same samples showed enrichment of Lindstedt dendritic cell maturation A (NES 2.16, FDR *q*-value < 0.0001), B (NES 2.20, FDR *q*-value < 0.0001), and C (NES 1.80, FDR *q*-value = 0.026) gene sets (Figure 5A). Furthermore, the leading-edge genes were those critical for adaptive immune activation, such as co-stimulatory molecules (Appendix A). Analysis of normalized RNA-seq data (cpm) had also shown up-regulation of co-stimulatory molecules (e.g., CD80, CD86, and CD40) and CD83 in fucoidan-treated BMDCs (Figure 5B). To complement our transcriptome analyses, we assessed the capacity of fucoidan to induce maturation of BMDCs by measuring the surface expression of DC maturation and T-cell co-stimulation markers by flow cytometry. After 24 h of treatment, MHCII, CD40, and CD80 were significantly upregulated (Figure 5C). 

## 4. Discussion

In the midst of a viral pandemic, fucoidan has been rapidly garnering interest for its antiviral activity [24,26]. Here, we analyzed global transcriptomic changes in fucoidan-treated BMDCs to better understand fucoidan’s antiviral properties. We found that fucoidan treatment upregulated viral-sensing receptors, their downstream signaling molecules, and subsequent effectors, effectively establishing an antiviral state in BMDCs.

Dendritic cells play a central role in the innate immune system to clear viral infections [29,34,64]. DCs recognize viral PAMPs and activate multiple signaling cascades, ultimately leading to the transcriptional induction of type I IFNs [6,29]. Ligand binding of type I IFN receptors will in turn activate the JAK-STAT pathway, driving the expression of hundreds of IFN-stimulated genes (ISGs) that directly and indirectly exert antiviral activities [6,65]. Both the DEGs and enriched gene sets in fucoidan-treated BMDCs showed strong enrichment of nucleic acid-sensing PRR signaling pathways such as that of RLR, TLR, NLR, and STING. Other enriched gene sets included that of inflammatory cytokines and chemokines, type I IFN production, and their respective receptors. Furthermore, fucoidan stimulation led to a strong induction of hundreds of ISGs as demonstrated by our Interferome analysis. These data are consistent with the general understanding that multiple PRR signaling pathways converge to activate inflammatory transcription factors such as NF-κB, IFN regulatory factor (IRF) 3, and IRF7 to subsequently induce cytokine, chemokine, and interferon (IFN) expression. As such, fucoidan may induce a self-enhancing, antiviral program in BMDCs, highlighting its clinical potential for suppressing virus activity.

In antiviral defense, DCs also act to activate and amplify the adaptive immune response [40,41,42]. DCs are professional antigen presenting cells (APCs) that capture viral pathogens for degradation and antigen presentation [41,49]. Naïve T cells recognize viral antigens only after processing and presentation by DCs in association with MHC molecules, leading to differentiation of antigen-specific Th1 CD4+ and CTL CD8+ T cells to eradicate the virus [48]. Our analysis of DEGs in fucoidan-treated BMDCs demonstrated strong induction of IL12, which is known to be a potent inducer of IFN-gamma production in NK cells to produce a Th1-skewed response [66,67]. Furthermore, GSEA and counts from our RNA-seq data showed that fucoidan treatment results in strong enrichment of dendritic cell maturation genes and co-stimulatory molecules, such CD80, CD86, CD40, and CD83, (Figure 5B). DEGs in fucoidan-treated BMDCs also included those related to DC activation, such as MHC I gene H2-K1, MHCI antigen presentation-related genes Tap1 and B2m, and MHC II genes H2-M3 and H2-Q4. These findings were further confirmed by our flow cytometry analysis showing that fucoidan upregulates surface expression of MHCII, CD80, and CD40 in BMDC cultures. Altogether, fucoidan may also enhance antigen presentation and co-stimulation functionality in BMDCs.

Given its ability to enhance both the innate and adaptive roles of DCs, fucoidan may effectively prevent and control a wide range of viral infectious diseases. While clinical observations during the recent outbreaks of coronaviruses have highlighted the importance of type I IFNs as a potent antiviral molecule, a state of hyper-inflammation has been linked to disease severity and poor outcomes following infection [68]. To appropriately manage the symptoms of COVID-19, it is crucial to balance adaptive immune activation for viral clearance, without leading to uncontrolled inflammation [68]. Interestingly, our results also suggest that fucoidan induces NRF2-based anti-inflammatory signaling pathways and heme oxygenase 1 (HO-1). It was recently demonstrated that several NRF2 activators suppress the inflammatory response to SARS-CoV-2 in human cells [69]. Upregulation of the NRF2-transcriptional target HO-1 (human gene name HMOX1) has also been linked to antiviral activity by catalyzing the degradation of heme into biliverdin, Fe^2+^, and CO, all of which have putative anti-SARS-CoV-2 activity [70]. As a potent inducer of both type I IFN signaling and NRF2 anti-inflammatory signaling, fucoidan-based nutraceuticals or functional foods may act as an effective, balanced strategy against COVID-19 [71].

The specific receptors responsible for fucoidan’s antiviral activity are not well known. As hydrophilic macromolecules, polysaccharides generally bind scavenger receptors (SRs), TLRs, complement receptor 3 (CR3), and C-type lectin receptors (CLRs) [62]. Sulfated polysaccharides in particular may bind multiple receptors due to their polyvalency. In our study, both CLR and TLR signaling pathways were among the KEGG pathways overrepresented in DEGs in fucoidan-treated BMDCs (Table 1). These were confirmed in our the subsequent GSEA, which additionally showed the enrichment of RLRs, CR3, class A SRs, and cytosolic DNA sensing molecules (Supp. Table 3). These findings are notable given that CLRs are known to route viral antigens such as Dectin1 to MHC class II compartments for presentation to CD4 + T cells [71]. In addition, SR-A has been shown to be a primary endocytic receptor that will internalize fucoidan and activate TLR3 and TLR9, thereby promoting type I IFN signaling [62,72]. It has been further reported that fucoidan may directly activate TLR-2 and TLR-4 [73]. Our findings are consistent with previous literature suggesting that TLRs, SRs, and CLRs may function as fucoidan receptors and subsequently lead to type I IFN signaling and ISG production.

In summary, we have demonstrated that the sulfated polysaccharide fucoidan induces global transcriptomic changes in BMDCs that may promote antiviral activity. Fucoidan-treated cells showed overrepresentation of virus-specific response pathways, including that of SARS-CoV-2, as well as gene sets related to nucleic acid-sensing PRR signaling, type I interferon (IFN) signaling, and ISG production. Our findings may serve as a key foundation for further research on the development of fucoidan-based strategies for the prevention and treatment of COVID-19 and other human viral diseases. In the future, critical observations obtained in the bioinformatic analyses need to be verified using other methods and also fucoidan will need to be investigated in animal models as an antiviral nutraceutical for boosting the type 1 interferon response to RNA viruses, such as influenza and coronaviruses.

## Figures and Tables

**Figure 1 nutrients-14-02242-f001:**
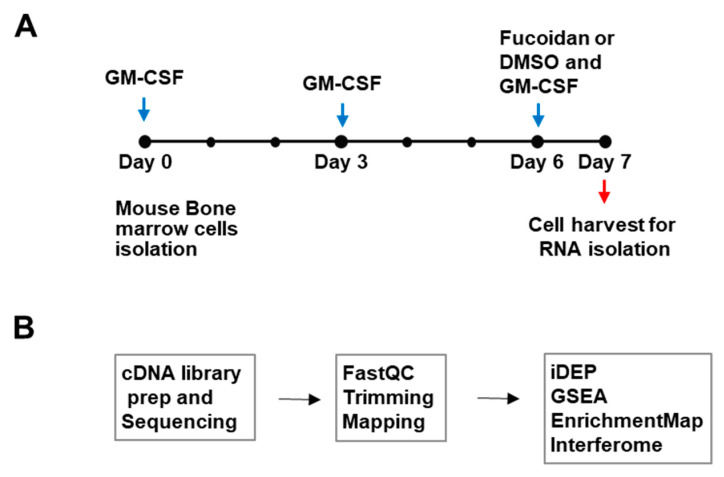
Outline of study design and transcriptome analysis. (**A**) Cell culture scheme. Mouse BMDCs were treated with Fucoidan or control (0.04% DMSO) and subjected to RNA-seq analysis (**B**) Transcriptomic analysis outline: cDNAs were generated, sequenced, and processed. Differentially expressed genes (DEGs) were identified and enriched pathways were analyzed with various bioinformatics tools.

**Figure 2 nutrients-14-02242-f002:**
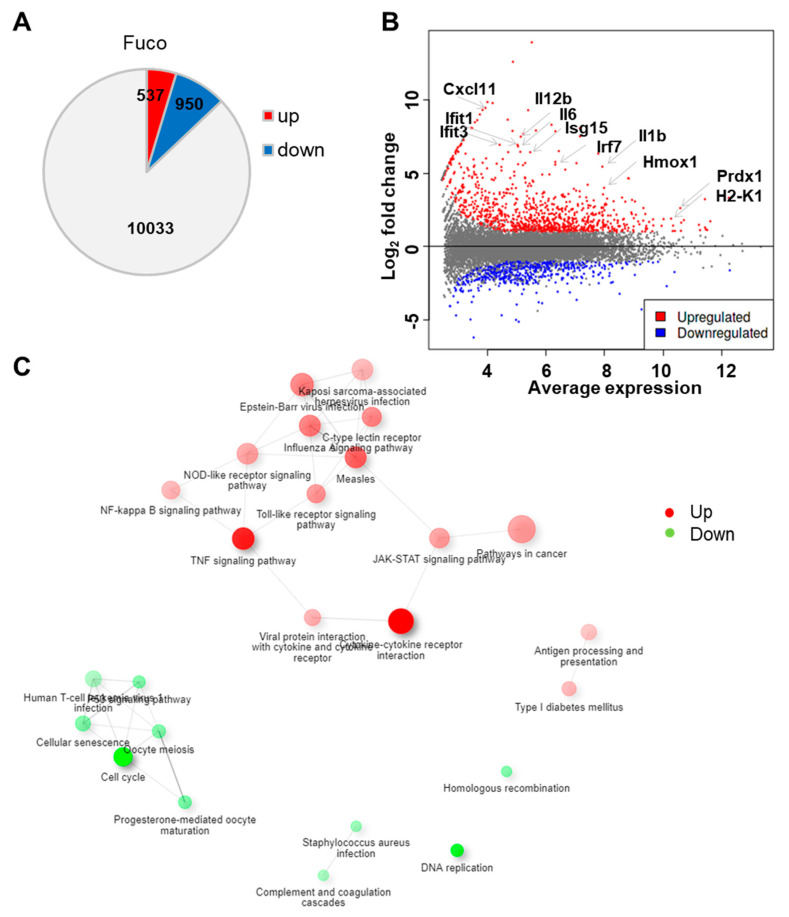
Fucoidan treatment upregulates genes related to immune response in BMDCs. (**A**) Number of differentially expressed genes in fucoidan-treated BMDCs relative to control-treated cells. (**B**) MA plot showing log fold-change and average gene expression in log count per million (cpm) in fucoidan-treated cells relative to control-treated cells. Representative upregulated genes are labeled with arrows. (**C**) Network visualization of overlaps among enriched KEGG pathways. Pathways are nodes and colored by the enrichment score, and edges are sized based on number of shared genes.

**Figure 3 nutrients-14-02242-f003:**
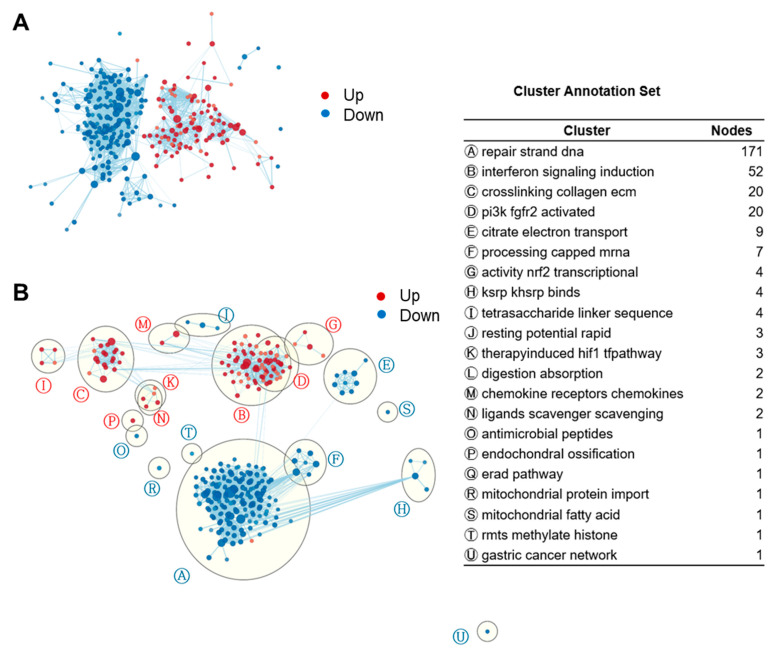
Enrichment maps of GSEA gene sets revealed strong induction of IFN signaling in fucoidan-treated BMDCs. (**A**) Unformatted GSEA gene sets enrichment map. Each node (circle) represents a distinct pathway, and edges (lines) represent the number of genes overlapping between two pathways, determined using the similarity coefficient. FDR < 0.08. Edge cutoff (Similarity) < 0.25. (**B**) AutoAnnotated enrichment map. Clusters were identified by the Cytoscape app and annotated with a representative label gleaned from the characteristics of the individual gene sets.

**Figure 4 nutrients-14-02242-f004:**
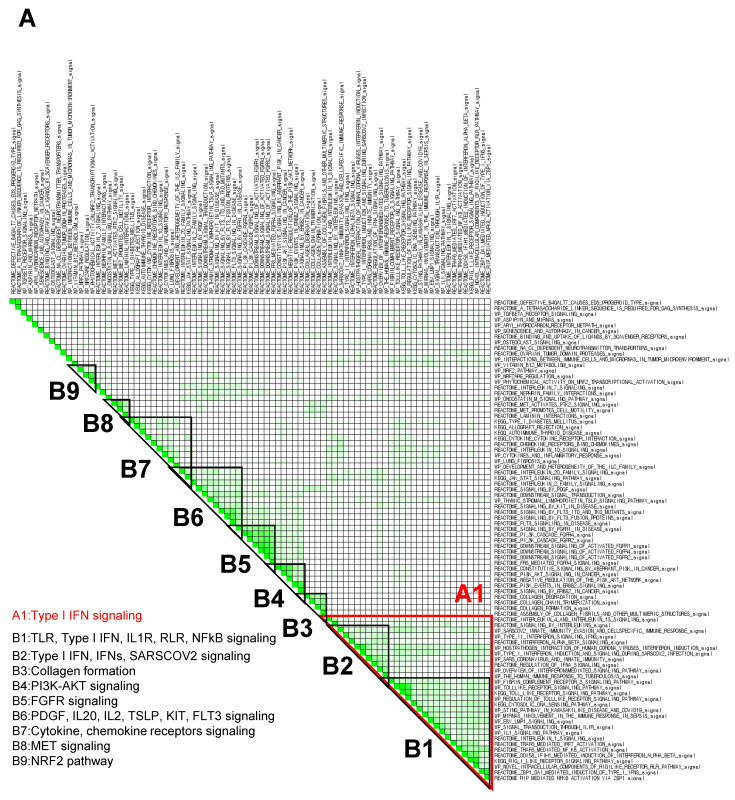
GSEA leading-edge analysis reveals that fucoidan activates a type I IFN response. (**A**) GSEA leading-edge analysis of significantly enriched gene sets from KEGG, Reactome, and WiKiPathways databases in the built-in C2 curated gene sets. The results are shown as a set-to-set diagram where the intensity of the green color directly correlates to the extent of the intersection between the leading-edge core genes of each gene set combination. The darker the color, the greater the overlap between subsets. (**B**) Representative GSEA enrichment plots from cluster A1 specifically related to type I IFN signaling and SARS-CoV-2 infection. False discovery rate (FDR) q value; normalized enrichment scores (NES). (**C**) Set-to-set diagram of GSEA based on transcription factor target database showing enrichment of STAT, IRF, NFkB target gene sets in fucoidan-treated BMDCs. (**D**) Venn diagram from Interferome analysis showing number of genes regulated by one or more IFN type (type I, II or III) among the 950 upregulated DEGs.

**Figure 5 nutrients-14-02242-f005:**
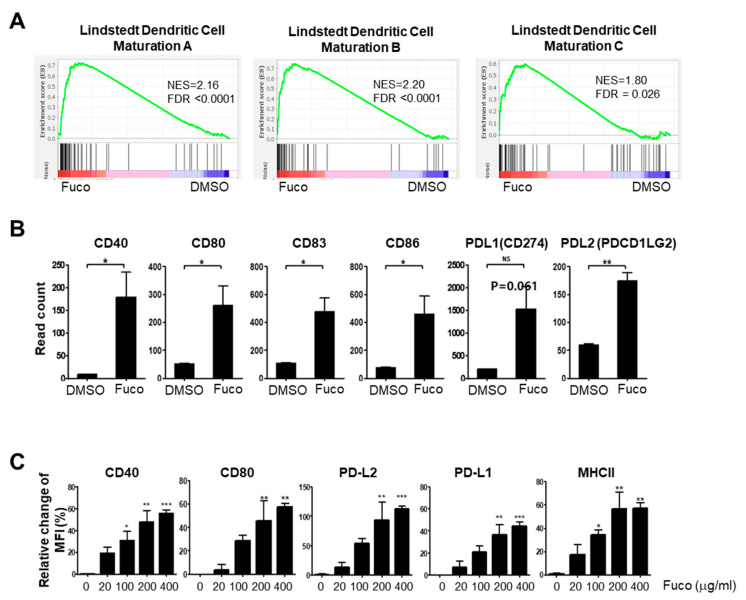
Fucoidan activates maturation of BMDCs. (**A**) Enrichment plots of Lindstedt DC maturation gene sets showing significant upregulation in fucoidan-treated BMDC. (**B**) Read counts of the DC maturation markers by RNA-seq. (**C**) Flow cytometry of DC maturation markers in fucoidan-treated BMDCs. Statistical difference by using the unpaired *t*-test relative to vehicle (**B**) or, one-way ANOVA with Dunnett’s post-test analysis compared to stained control (0 μg/mL) (**C**). The bar above the data displays ±SEM. Asterisks depict levels of significance as follows: NS, not significant (*p* ≥ 0.05); * *p* < 0.05; ** *p* < 0.01; *** *p* < 0.001.

**Table 1 nutrients-14-02242-t001:** Enriched KEGG pathways among genes up- and down-regulated by fucoidan treatment.

Direction	Pathways	nGenes	adj.Pval
Down	Cell cycle	24	8.90 × 10^−9^
Regulated	DNA replication	11	4.40 × 10^−6^
	Progesterone-mediated oocyte maturation	12	1.70 × 10^−3^
	Homologous recombination	8	2.10 × 10^−3^
	P53 signaling pathway	11	2.10 × 10^−3^
	Cellular senescence	17	2.10 × 10^−3^
	Oocyte meiosis	13	2.10 × 10^−3^
	Staphylococcus aureus infection	8	4.30 × 10^−3^
	Complement and coagulation cascades	8	6.50 × 10^−3^
	Human T-cell leukemia virus 1 infection	19	8.60 × 10^−3^
Up	Cytokine–cytokine receptor interaction	54	4.80 × 10^−20^
Regulated	TNF signaling pathway	39	5.70 × 10^−16^
	Measles	36	5.00 × 10^−12^
	Influenza A	36	1.10 × 10^−10^
	Epstein–Barr virus infection	42	2.30 × 10^−10^
	C-type lectin receptor signaling pathway	29	4.30 × 10^−10^
	Toll-like receptor signaling pathway	26	1.90 × 10^−9^
	JAK-STAT signaling pathway	31	6.70 × 10^−9^
	Pathways in cancer	67	6.70 × 10^−9^
	NOD-like receptor signaling pathway	35	1.20 × 10^−8^
	Viral protein interaction with cytokine and cytokine receptor	20	3.90 × 10^−8^
	NF-kappa B signaling pathway	25	7.70 × 10^−8^
	Type I diabetes mellitus	15	7.70 × 10^−8^
	Kaposi sarcoma-associated herpesvirus infection	35	9.60 × 10^−8^
	Antigen processing and presentation	18	3.30 × 10^−7^

**Table 2 nutrients-14-02242-t002:** Transcription factor pathways with target genes upregulated by fucoidan treatment.

Pathways	Genes
Irf7 target gene	Isg15 B2m Cd80 H2-K1 H2-M3 H2-Q4 Irf8 Irgm1 Cxcl10 Ifit1 Ifit2 Ifit3 Igtp Ikbkg Irf1 Irf4 Irf9 Tlr7 Gbp4 Mt2 Pml Lgals3bp Trim30a Ccl5 Spp1 Stat1 Tank Tap1 Tnf Traf6 Cmpk2 Oasl1 Ddx60 Oasl2 Usp18 Parp12 Oas3 Xaf1 Parp14 Rsad2 Zbp1 Iigp1 Rtp4 Bst2 Ifi35 Ifih1 Uba7 Dhx58 Ifi44
Nfkb1 target gene	Acp5 Birc3 Cxcr5 Casp1 Cd14 Cebpb Cflar Socs3 Csf1 Cxcl1 Cfb Hif1a Cxcl10 Ikbkg Il12a Il12b Il15 Il1a Il1b Il1rn Il2ra Il6 Kdr Gadd45b Nfkb1 Nfkb2 Nfkbia Nfkbib Nfkbie ENSMUSG00000023947 Nos2 Sqstm1 Eif2ak2 Ptgs2 Ripk2 Rel Ccl3 Ccl4 Ccl5 Cxcl2 Sdc4 Tapbp Tnf Tnfaip3 Traf1 Traf6 Vegfa Akt3 Malt1 Map3k8 Tnfsf15 Ebi3 Map3k14 Zbp1 Nfkbiz Il23a
Stat1 target gene	Isg15 Ahr Apbb2 Rhoc Atf3 B2m Bmpr2 Cacna1d Casp1 Casp4 Ccnd2 Socs3 Socs1 Crem Fosl1 H2-K1 H2-M3 H2-Q4 Irf8 Irgm1 Cxcl10 Ifit1 Ifit2 Ifit3 Il18bp Igtp Il2ra Il6 Irf1 Irf4 Irf9 Jak3 Lta Hook2 Man1a Mt1 Mt2 Nos2 Osm Enpp2 Pik3r1 Pml Lgals3bp Eif2ak2 Dusp1 Rab10 Trim30a Rras Ccl2 Sphk1 Trim21 Stat1 Stat3 Stat5a Tap1 Socs2 Tnf Tnfrsf8 Traf6 Cmpk2 Ubc Vdr Vegfa Yes1 Ikzf4 Gbp5 Oasl1 Ppp2r3a Oasl2 Usp18 Parp12 Vasn Mllt6 Oas3 Etv3 Dock6 Pik3r5 Txndc17 Map3k14 Irf7 Parp14 Gbp4 Mtor Rsad2 Zbp1 Iigp1 Cd274 Asph Herc6 Cpeb4 Rtp4 Ifi35 Ifih1 Ppme1 Uba7 Dtx2 Ddit4 Ccdc6 Usp6nl Ifi44
Irf9 target gene	Isg15 B2m H2-K1 H2-M3 H2-Q4 Irf8 Irgm1 Ifit1 Ifit3 Igtp Irf1 Irf4 Mt2 Pml Stat1 Stat2 Stat3 Stat4 Stat5a Oasl1 Oasl2 Usp18 Oas3 Irf7 Parp14 Rsad2 Rtp4 Ifi35
Irf5 target gene	B2m H2-K1 H2-M3 H2-Q4 Irf8 Il12a Il12b Il1b Il6 Irf1 Irf4 Irf9 Tlr7 Mt2 Pml Ccl3 Ccl4 Ccl5 Stat1 Tnf Traf6 Ubc Oasl1 Oas3 Irf7
Rela target gene	Spred2 Ahr Birc3 Casp1 Cd14 Cebpb Socs3 Socs1 Csf3 Igf2bp1 F3 Cxcl1 Hif1a Ier3 Cxcl10 Ikbkg Il12a Il12b Il1a Il1b Il2ra Il6 Maff Gadd45b Nfkb1 Nfkb2 Nfkbia Nfkbib Nfkbie ENSMUSG00000023947 Nos2 Nr4a2 Sqstm1 Ptgs2 Ripk2 Rel Ccl2 Ccl3 Ccl4 Ccl5 Cxcl2 Sema4c Socs2 Tnf Tnfaip3 Tnfrsf1b Tnfrsf9 Traf1 Traf6 Ubc Vegfa Akt3 Malt1 Mllt6 Tnfsf15 Ebi3 Map3k14 Zbp1 Rffl Nfkbiz Il23a
Irf1 target gene	Isg15 Spred2 Agrn B2m Cdkn2b F3 Fmn1 H2-K1 H2-M3 H2-Q4 Irf8 Irgm1 Cxcl10 Igtp Il12a Il12b Il15ra Il6 Irf4 Irf9 Mt2 Osm Pik3r1 Pml Lgals3bp Eif2ak2 PSME2b Stat1 Stat3 Stat4 Dtx3l Ncoa7 Tap1 Tapbp Tgif1 Oasl1 Ets2 Usp18 Oas3 Tmtc2 Ccbe1 Flrt2 C1ra Irf7 Pcdh7 Parp14 Gbp4 Rybp Chst11 Cd274 Cxcl16 Grina Rtp4 Tmem140 Bst2 Ifi35 Znfx1
Irf8 target gene	B2m H2-K1 H2-M3 H2-Q4 Il12a Il12b Irf1 Irf4 Irf9 Mt2 Pml Ccl5 Stat1 Traf6 Ubc Oasl1 Oas3 Irf7
Relb target gene	Ahr Bcl3 Cdkn1a Daxx Cxcl1 Ikbkg Nfkb1 Nfkb2 Nfkbia Nfkbib Nfkbie ENSMUSG00000023947 Cxcl2 Tnf Cd40 Map3k14
Irf6 target gene	B2m H2-K1 H2-M3 H2-Q4 Irf8 Irf1 Irf4 Irf9 Mt2 Pml Stat1 Oasl1 Oas3 Irf7
Pml target gene	B2m Daxx H2-K1 H2-M3 H2-Q4 Irf8 Irf1 Irf4 Irf9 Mt2 Pml Skil Stat1 Ubc Oasl1 Oas3 Irf7 Mtor
Nfkb2 target gene	Birc3 Bcl3 Casp1 Ikbkg Il1b Nfkb1 Nfkbia Nfkbib Nfkbie ENSMUSG00000023947 Nos2 Ptgs2 Tnf Traf1 Traf6 Map3k14 Zbp1 Nfkbiz
Irf2 target gene	Isg15 Spred2 Agrn Fabp4 B2m Cdkn2b F3 Fmn1 H2-K1 H2-M3 H2-Q4 Irf8 Cxcl10 Il15ra Irf1 Irf4 Irf9 Mt2 Osm Pik3r1 Pml Lgals3bp PSME2b Stat1 Ncoa7 Tap1 Tapbp Suco Oasl1 Usp18 Oas3 Tmtc2 Rabgap1l Ccbe1 Irf7 Pcdh7 Rybp Chst11 Cxcl16 Grina Rtp4 Bst2 Ifi35 Ogfr Mvp Znfx1
Tnfaip3 target gene	Ikbkg Nfkb1 Nfkbia Nfkbie ENSMUSG00000023947 Ripk2 Tnf Tnfaip3 Traf1 Traf6 Tnip3 Tax1bp1 Tnip1 Nfkbiz
Ikbkg target gene	Erc1 Birc3 Hspa1b Ikbkg Nfkb1 Nfkb2 Nfkbia Nfkbib Nfkbie ENSMUSG00000023947 Tank Tnfaip3 Cd40 Traf1 Traf6 Ubc Akt3 Malt1 Map3k8 Rnf31 Map3k14 Irf7 Zbp1 Nfkbiz

**Table 3 nutrients-14-02242-t003:** Gene sets in cluster A1 (Type I IFN signaling).

NAME	SIZE	NES	NOM *p*-Value
REACTOME_INTERFERON_ALPHA_BETA_SIGNALING	58	2.028	0.000
WP_CYTOKINES_AND_INFLAMMATORY_RESPONSE	24	1.933	0.000
WP_SARSCOV2_INNATE_IMMUNITY_EVASION_AND_CELLSPECIFIC_IMMUNE_RESPONSE	62	1.925	0.000
WP_TYPE_I_INTERFERON_INDUCTION_AND_SIGNALING_DURING_SARSCOV2_INFECTION	28	1.838	0.000
WP_OVERVIEW_OF_INTERFERONSMEDIATED_SIGNALING_PATHWAY	30	1.829	0.000
WP_SARS_CORONAVIRUS_AND_INNATE_IMMUNITY	27	1.825	0.000
WP_TYPE_II_INTERFERON_SIGNALING_IFNG	31	1.821	0.000
WP_EBV_LMP1_SIGNALING	23	1.816	0.005
WP_HOSTPATHOGEN_INTERACTION_OF_HUMAN_CORONA_VIRUSES_INTERFERON_INDUCTION	32	1.805	0.003
KEGG_RIG_I_LIKE_RECEPTOR_SIGNALING_PATHWAY	63	1.795	0.000
REACTOME_DDX58_IFIH1_MEDIATED_INDUCTION_OF_INTERFERON_ALPHA_BETA	72	1.767	0.000
WP_TOLLLIKE_RECEPTOR_SIGNALING_PATHWAY	99	1.766	0.000
KEGG_TOLL_LIKE_RECEPTOR_SIGNALING_PATHWAY	97	1.763	0.000
REACTOME_TRAF6_MEDIATED_NF_KB_ACTIVATION	23	1.756	0.005
REACTOME_REGULATION_OF_IFNA_SIGNALING	22	1.755	0.000
REACTOME_TRAF6_MEDIATED_IRF7_ACTIVATION	25	1.737	0.005
REACTOME_INTERLEUKIN_4_AND_INTERLEUKIN_13_SIGNALING	101	1.724	0.000
REACTOME_SIGNALING_BY_INTERLEUKINS	422	1.720	0.000
WP_MIRNAS_INVOLVEMENT_IN_THE_IMMUNE_RESPONSE_IN_SEPSIS	53	1.713	0.000
KEGG_CYTOSOLIC_DNA_SENSING_PATHWAY	49	1.660	0.003
WP_REGULATION_OF_TOLLLIKE_RECEPTOR_SIGNALING_PATHWAY	137	1.655	0.000
REACTOME_ZBP1_DAI_MEDIATED_INDUCTION_OF_TYPE_I_IFNS	20	1.637	0.019
REACTOME_INTERLEUKIN_1_SIGNALING	99	1.618	0.000
WP_NOVEL_INTRACELLULAR_COMPONENTS_OF_RIGILIKE_RECEPTOR_RLR_PATHWAY	57	1.608	0.000
WP_STING_PATHWAY_IN_KAWASAKILIKE_DISEASE_AND_COVID19	18	1.606	0.010
WP_THE_HUMAN_IMMUNE_RESPONSE_TO_TUBERCULOSIS	22	1.605	0.016
WP_SIGNAL_TRANSDUCTION_THROUGH_IL1R	34	1.603	0.011
WP_IL1_SIGNALING_PATHWAY	56	1.595	0.000
REACTOME_RIP_MEDIATED_NFKB_ACTIVATION_VIA_ZBP1	17	1.565	0.026

**Table 4 nutrients-14-02242-t004:** Leading-edge genes shared by type I IFN signaling pathways in cluster A1.

Number of Gene Sets	Gene
21	Nfkb1
19	Ikbkb, Ikbkg, Nfkbia
18	Ifnb1, Traf6
15	Tbk1
14	Stat1
12	Ikbke, Tnf, Mapk8
11	Il1b, Ifnar2, Ifnar1, Nfkb1b, Irf7
10	Stat2, Cxcl10, Il6
9	Nfkb2, Ddx58
8	Jun, Ticami
7	Il12b, Pi3kr1, Tollip, Ifih1, Tlr7, Tlr3, Socs1, Irf9, Ccl5, Ccl4
6	Trim25, Il12a, Nkiras1, Il1a, Map3k1, Map2k1, Irak2, Traf2, Pik3ca, Ccl3
5	Jak2, Jak1, Tank, Saa1, Map3k8, Socs3, Isg15, Cxcl9, Irak3, Ccl2
4	App, Icam1, Dhx58, Il1r1, Cd14, Ifna4, Il10, Peli1, Tlr4, Psmb8, Cd86, Cd80, Irf1, Nos2, Il1rap, Sqstm1, Ly96, Cxcl11, Pik3cb, Hsp90aa1, Irf4, Traf3
3	Il1rn, Ifit2, Akt3, Atg12, Pik3r5, Eif2ak2, Cyld, Rel, Cd40, Syk, Ifngr2, Tlr1, Tlr6, Spp1, Psmb9, Zbp1, Ube2v1, Ubc, Map3k14, Casp8, Oas2, Oas3

## Data Availability

All raw and processed sequencing data generated in this study have been submitted to the NCBI Gene Expression Omnibus (GEO; https://www.ncbi.nlm.nih.gov/geo/) under accession number GSE200944 and the Korean Nucleotide Archive (KoNA; https://kobic.re.kr/kona) under KoNA access ID PRJKA220176.

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
