# Peer review of "Gene Set Enrichment Analysis Reveals That Fucoidan Induces Type I IFN Pathways in BMDC"

_nutrients, 2022, doi:10.3390/nu14112242_

Round 1

Reviewer 1 Report

In order to explor the mechanisms on anti-viral activity of fucoidan, this study invetigated the global transcriptional changes of BMDCs with RNA-Seq technology, and results indicated that fucoidan transcriptionally activates PRR, type I IFN, and adaptive immune stimulatory signals in DCs to ultimately enhance the antiviral activity of dendritic cells. After careful review, I think this article is informative and acceptable after condsidering the data is reliable and conclusions are credible. A small problem is that the tables are nots clear enough, it is better to convert them from picture style to table form.

Reviewer 2 Report

This is an interesting study and the authors have collected a unique dataset about the mechanisms behind fucoidans' anti-viral activity. The paper is generally well written and structured. There are some minor issues that should be addressed as recommended follow:

1-The abstract is nicely written, however, does not reveal the depth of information presented in the manuscript.

2-Typos and grammatical mistakes should be corrected before publication. (for example: mg/ml)

3-The authors should discuss more future directions.

Reviewer 3 Report

General comments

The manuscript by Suyoung Choi et al. studied fucoidan induced changes in BMDC at the transcriptional level. DEG analysis revealed upregulated gene sets and pathways involved in immune function, such as pathogen recognition, antigen processing and presentation. GSEA analysis identified genes enrichments in IFN signaling, mainly type I IFN response. Flow cytometric analysis showed enhanced antigen presentation and co-stimulation functionality. There is much new and useful information in here, however minor revision is still needed.

Specific comments

  1. In “2.1.”, BMDCs were treated by 400 μg/ml fucoidan, please provide the basis. Also, please add physical and chemical characteristics for this commercial fucoidan.  
  2. 2. In flow cytometry, BMDCs were stained with a panelof antibodies against with antibody for cell surface marker, but not antibodies against type I IFN cytokines, please justify.
  3. The major weakness with this work is lacking in verification experiments for the results of DEG and GSEA analysis using either qPCR or western blot, which will essentially make this work being interesting.
